# Obsessive Compulsive Symptoms and Psychopathological Profile in Children and Adolescents with KBG Syndrome

**DOI:** 10.3390/brainsci9110313

**Published:** 2019-11-07

**Authors:** Paolo Alfieri, Francesco Demaria, Serena Licchelli, Ornella Santonastaso, Cristina Caciolo, Maria Cristina Digilio, Lorenzo Sinibaldi, Chiara Leoni, Maria Gnazzo, Marco Tartaglia, Patrizio Pasqualetti, Stefano Vicari

**Affiliations:** 1Department of Neuroscience, Child and Adolescent Psychiatry Unit, Bambino Gesù Children’s Hospital, IRCCS, 00165 Rome, Italy; francesco.demaria@opbg.net (F.D.); serena.licchelli@gmail.com (S.L.); ornella.santonastaso@opbg.net (O.S.); cristina.caciolo@opbg.net (C.C.); stefano.vicari@opbg.net (S.V.); 2Genetics and Rare Diseases Research Division, Bambino Gesù Children’s Hospital, IRCCS, 00165 Rome, Italy; mcristina.digilio@opbg.net (M.C.D.); lorenzo.sinibaldi@opbg.net (L.S.); maria.gnazzo@opbg.net (M.G.); marco.tartaglia@opbg.net (M.T.); 3Center for Rare Diseases and Congenital Defects, Fondazione Policlinico Universitario A. Gemelli, IRCCS, 00168 Rome, Italy; leonichia2@gmail.com; 4Institute of Pediatrics, Università Cattolica del Sacro Cuore, 00168 Rome, Italy; 5Fatebenefratelli Foundation for Health Research and Education, San Giovanni Calibita-Fatebenefratelli Hospital, Isola Tiberina, 00186 Rome, Italy; patrizio.pasqualetti@afar.it; 6Institute of Psychiatry, Fondazione Policlinico Universitario A. Gemelli, Università Cattolica del Sacro Cuore, 00168 Rome, Italy

**Keywords:** obsessive compulsive symptoms, developmental disorders, *ANKRD11*

## Abstract

KBG syndrome is a rare multisystem developmental disorder caused by ankyrin repeat domain-containing protein 11 (ANKRD11) gene haploinsufficiency, resulting from either intragenic loss-of-function mutations or microdeletions encompassing the gene. Concerning the behavioral phenotype, a limited amount of research has been focused on attention deficit and hyperactivity disorder, autistic-like features, anxiety and impairments in emotion regulation, and no study has provided a systematic assessment. The aim of the present work is to investigate the psychopathological profile in children, adolescents, and young adults with KBG syndrome. Seventeen subjects with molecularly confirmed diagnoses were evaluated to investigate cognitive abilities and psychopathological features. Parametric and nonparametric indexes were used to describe the patient cohort according to type and distribution of specific measures. The KBG subjects were characterized by a low mean IQ score, with a distribution characterized by a variability similar to that occurring in the general population. Prevalence of neuropsychiatric disorders were computed as well as the corresponding confidence intervals to compare their prevalence to that reported for the general population. The KBG subjects were characterized by higher prevalence of obsessive-compulsive, tic, depressive and attention deficit and hyperactivity disorders. Obsessive-compulsive disorder is a peculiar aspect characterizing the psychopathological profile of KBG patients, which does not seem to be related to the cognitive level. The present study provides new relevant information towards the definition of a psychopathological phenotype of KBG syndromes useful to plan a better treatment for patients.

## 1. Introduction

KBG syndrome (MIM #148050) is a rare multisystemic developmental disorder [1,2], the acronym deriving from the surname initials (K, B and G) of the first three families diagnosed with the disorder [3]. Approximately 180 patients have been reported worldwide [4,5,6,7,8]. KBG syndrome is caused by ankyrin repeat domain-containing protein 11 (ANKRD11) gene haploinsufficiency, resulting from either loss-of-function intragenic mutations or 16q24.3 chromosome microdeletions in encompassing the gene [4,6,9,10,11,12,13,14]. Although clinical features may vary, the core symptoms of KBG syndrome include developmental delay (DD)/intellectual disability (ID), and a distinctive gestalt (triangular face, brachycephaly, hypertelorism, protruding ears, an upturned nose with full nasal tip, and macrodontia) [15,16]. In addition, KBG syndrome is frequently associated with hearing problems, short stature as well as cardiac defects, palate abnormalities, sleep disturbances, abnormal electroencephalogram (EEG) findings, and feeding difficulties during infancy [4,7,8,17].

Recently, Ka and Kim highlighted the role of ANKRD11 in regulating pyramidal neuron migration and dendritic differentiation in the developing mouse cerebral cortex [1]. This finding has provided insights into the molecular mechanism underlying the neurodevelopmental features associated with defective ANKRD11 function. Cognitive skills in KBG vary among individuals. Most patients have some degree of DD, especially in speech, even though verbal intelligence quotient (IQ) can be scored as higher compared to performance IQ in adulthood [6,7,8,9,10,14,15,16,17,18]. The cognitive profile can range from mild learning disabilities, particularly in females, to moderate ID, which more commonly occur in males, with a majority of patients demonstrating mild ID [10,15]. While affected subjects rarely complete regular high school without support, more than half of the affected adults have jobs and are self-sufficient [7,8].

Longitudinal studies in subjects with ID have documented that lower childhood IQ may be a risk factor for the development of psychiatric disorders during adulthood [19]. Recently, bidirectional causal relationships between psychopathological disorders and IQ level showed that elevated disorder-specific severity is associated to lowered IQ in the general pediatric population [20].

The psychopathological features of KBG syndrome have been anecdotally described to date; the few reports, included case studies, mainly reported attention deficit and hyperactivity disorder (ADHD) and autistic-like features in both children and adults [4,7,8].

While ADHD has been clearly associated with KBG syndrome, the link with autism spectrum disorders (ASD) is debated [7,8]. The association of ASD with the 16q24.3 deletion including *ANKRD11* [13,21], and with the intragenic *ANKRD11* pathogenic variants strongly varies in the different cohorts of patients reported to date, leading to hypothesize the presence of ascertainment biases [5,7,22].

Anxiety and shyness have been reported to be associated with difficulties in understanding social situations and making friendships; some patients have obsessions and can fix themselves on a favorite object or a routine or obnoxious change with a behavior that deteriorates during times of stress. Moreover, difficulties in regulating emotions with outbursts of anger, inconsolable disturbance and occasional aggressive explosions have also been described [8]. A recent study investigating behavioral aspects by using questionnaires and observations in 18 patients with KBG syndrome in comparison to 17 patients with other genetic disorders matched for intelligence levels showed impulsivity, restless behavior, distractibility, and impairments in emotion regulation as more prevalent in the former [18].

Although several studies have highlighted psychopathological problems in patients with KBG syndrome, there are no studies providing a systematic assessment of these aspects. Based on these considerations, the aim of the present study was to profile the psychopathological features in children and adolescents with a molecularly confirmed diagnosis of KBG syndrome and explore the occurrence of a possible association with IQ level.

## 2. Materials and Methods

Seventeen subjects were included in this study. All subjects participated voluntarily. Informed consent was obtained from all parents prior to participation and after receiving a comprehensive description of the study. The study was performed in accordance with the Declaration of Helsinki (1964) and was approved by the local ethical committee of the Bambino Gesù Children’s Hospital.

In all participating subjects, the clinical assessment documented features fitting the diagnosis of KBG syndrome, which was confirmed by molecular analysis showing de novo microdeletions or mutations of the *ANKDR11* gene in all cases except one with frameshift mutation transmitted from an affected mother (Table 1 and Table 2). All patients were regularly followed at the Medical Genetics Unit of the Bambino Gesù Children’s Hospital and Center for Rare Diseases and Congenital Defects of Gemelli Hospital. Independently from the presence of psychopathological problems, the patients were evaluated by experienced clinicians at the Child and Adolescent Psychiatry Unit of the Bambino Gesù Children’s Hospital with the only criteria being the presence of a molecularly confirmed diagnosis of KBG. Twelve out of 16 patients attend school with support teaching, while an adult patient was involved in a rehabilitation program to support inclusion and living skills (daily frequency, in a flat with other 5 individuals). Three patients were treated by single antiepileptic medication (2 patients by levetiracetam, 1 by carbamazapine) and the adult patient had been treated by valproic acid during childhood. Participants were evaluated by an expert child psychiatrist investigating psychopathological signs/psychiatric disorders according to criteria of Diagnostic and Statistical Manual of Mental Disorders fourth edition (DSM-IV-TR) [23], and assessed with rating scales by experienced psychologists. General cognitive abilities were scored with age-scaled tests based on age, language and cognitive skills, including the Wechsler Intelligence Scale for Children-Fourth Edition (WISC-IV) [24] and, for two individuals, the Italian version of the Wechsler Adult Intelligence Scale-Fourth Edition (WAIS-IV) [25]. The Leiter International Performance Scale, third edition [26], was administered for two cases showing language impairment and ID.

Children and parents were interviewed by using the Schedule for Affective Disorders and Schizophrenia for School Age Children, Present and Lifetime version (KSADS PL) [27] in order to detect current and past features of psychopathological signs/psychiatric disorders according to DSM-IV-TR criteria.

The Children’s Global Assessment Scale (CGAS) [28] was used to estimate the children’s current level of functioning (see the Appendix A for details).

Psychopathological aspects were evaluated using parent and self-report questionnaires: Child Behavior Checklist for Ages 6–18 (CBCL/6–18) [29], Multidimensional Anxiety Scale for Children (MASC) [30], and the Children’s Depression Inventory (CDI) [31] (see the Appendix A).

The presence of a symptomatology characterized by obsessions and compulsions was evaluated through the administration of the Children’s Yale–Brown Obsessive-Compulsive Scale (CY–BOCS) [32] (see Appendix A).

Parametric (mean, SD) and nonparametric (median, min–max) indexes were used to describe the sample of KBG patients, according to type and distribution of specific measures.

Confidence intervals of prevalence estimates were computed on the basis of Wilson’s procedure. This procedure is particularly appropriate for small sample sizes and should be considered a better approach with respect to exact binomial procedure. This approach can be substantiated on the grounds that it is the exact algebraic counterpart to the large-sample hypothesis test and was recommended by Agresti and Coull for virtually all combinations of *n* and *p*, even when the occurrence of an event was 0. Thus, it allowed the comparisons of KBG prevalence of various traits/disorders to that observed in the general population (GP).

## 3. Results

Descriptive statistics are reported in Table 1.

The study sample comprised 9 males (53%) and 8 females (47%), aged between 7.3 and 23.9 years with clinical and genetic diagnoses of KBG. Most of patients showed a truncating frameshift (*n* = 12, 71%) or nonsense (*n* = 4, 23%) variants, while one subject (6%) carried a deletion of the entire gene (Table 2).

KBG subjects were characterized by a low mean IQ score (M = 66.0, SD = 16.2), with a distribution characterized by a similar variability compared to the general population (SD = 15).

As regard Children’s Yale–Brown Obsessive-Compulsive Scale (CY-BOCS), the total score was 10 (median), ranging from 0 to 30. CY-BOCS Compulsion was slightly increased compared to CY-BOCS Obsession (Wilcoxon, *p* = 0.059).

The CGAS score mean was 58.7 (SD = 8.8), ranging from 50 to 75.

The MASC-total score (mean = 51.3, SD = 13.8) and MASC-Anxiety Disorder Index (mean = 48.6, SD = 12.7) were very close to the GP expected values (mean = 50, SD = 10). However, the MASC profile in the KBG cohort was not flat ((F(2.6;25.8) = 4.756; *p* = 0.012), since the score of MASC-separation subscale (60.5, SD = 11.4) was significantly higher (*p* < 0.05, consistently) than the other subscales (physical symptoms: 49.1, SD = 13.5; social anxiety: 50.0, SD = 12.5; harm-avoidance: 45.0; SD = 12.4), where means were closer to GP expected values.

The CBCL/6–18 data of our cohort were reported in Table 3. In the last two columns of the table, the percentages of our cohort that obtain clinical or borderline scores in each subscale of CBCL/6–18 are reported. Specifically, according to ASEBA Assessment Data Manager (ADM) the scored profile of Syndrome Scales and DSM-Oriented Scale, *t*-scores from 67 to 70 are in the borderline range and *t*-scores above 70 are in the clinical range; concerning the Total Problem, Internalizing, and Externalizing Scale, *t*-scores of 60 to 63 delineate the borderline range, while *t*-scores above 63 delineate the clinical range [29].

For each item of KSADS-PL, prevalence of trait, disorder and trait-or-disorder were computed as well as the corresponding confidence intervals, in order to compare their prevalence to what is reported in the GP. These findings are reported in Table 4. The KBG subjects showed higher prevalence of obsessive-compulsive disorder (OCD) traits or disorder (82% vs. 7.1% of GP, *p* < 0.001) [33], ADHD disorder (no patients in our cohort with traits) (29% vs. 9.9% of GP, *p* < 0.007) [34], tic disorders (17.6% vs. 6.7% of GP, *p* < 0.073) [35], and depressive traits or disorder (24% vs. 8.2% of GP, *p* < 0.001) [36] compared to the age-matched unaffected population.

Table 5 reports the prevalence of specific OCD behavioral symptoms ranked from highest (hoarding compulsion, 65%) to lowest. In addition, the number of obsessive/compulsive behaviors was computed with the corresponding Poisson confidence intervals, indicating that in a KBG subject less than 1 obsessive behavior could be expected, whereas more than 1 compulsive behavior were diagnosed. This dominance of compulsive behaviors was confirmed by the specific index (theoretically ranging from −1 and 1), whose value was 0.38 (95% CI: 0.05, 0.70; *p* = 0.034 vs. symmetry). This finding confirms higher CY-BOCS scores for compulsive vs. obsessive domain.

## 4. Discussion

Herein, we examined the prevalence of psychopathological features, IQ, and global functioning in a cohort of 17 children and adolescents with molecularly confirmed KBG syndrome. To our knowledge, this is the first study that systematically assessed a comprehensive spectrum of psychopathological features in young patients with unambiguous clinical assignment for this rare disease. In our sample, none of the patients met criteria for ASD and only 12% of cases in our cohort showed positivity to the “social problem” subscale of CBCL/6–18 in, confirming data by a recent study describing autism in KBG as an aspect less frequent than previously reported [7].

Our data confirmed a high prevalence of OCD or traits of disorder in the KBG syndrome, particularly if compared to the age-matched unaffected population. Most of the investigated individuals showed OCD features, with one-third meeting DSM-IV-TR criteria for OCD disorder, and more than half showing subclinical traits.

Investigating obsessive compulsive symptomatology, we found higher prevalence of compulsions compared to obsessions; in particular, we noted several compulsion behaviors, the most common one was compulsive hoarding, slightly less compulsive ordering, and compulsive checking, while more classical compulsive washing was of relatively low frequency (Table 5). For all patients in whom OCD symptoms were identified both patients and parents described characteristics of finalized behaviors (to prevent or reduce anxiety or distress, or prevent some dreaded event or situation). What is worthy of note is that despite none of the patients meeting the criteria for ASD, mostly in cases of low IQ, ASD features and OCD compulsions can be difficult to distinguish and this may represent risk of misdiagnoses and missed diagnoses.

We demonstrated an association between the severity of obsessive-compulsive symptomatology, specifically for compulsions, with a worse global functioning, as measured by CGAS. This is in line with previous reports on other genetic syndromes such as 22q11.2 deletion syndrome (DS) and the Prader–Willi syndrome [37,38]. Even though chromosomal rearrangements have been reported in a small number of individuals with OCD, no susceptibility genes in OCD have been identified yet [39]. Like in other neuropsychiatric conditions, the challenge of responsible genes identification could be due to the clinical and genetic heterogeneity of the disorder.

Utilizing large cohorts, some longitudinal studies have concluded that lower childhood IQ may be a risk factor for the development of psychiatric disorders later in life [19]. However, a recent large study demonstrated bidirectional causal relationships between psychopathology problems and IQ level [20]. Keyes and colleagues examined a sample of 10.073 American adolescents for which IQ and psychopathology information was available and found that lifetime psychopathology was generally associated with lower IQ [20]. Notwithstanding, the authors found that among individuals diagnosed with psychiatric disorders, elevated disorder-specific severity was associated with lowered IQ as well. The present study also confirms the heterogeneity on IQ scores observed in patients with KBG syndrome, although the variability in our sample seems to be similar to what has been reported in the general population. There was no correlation between IQ scores and obsessive-compulsive symptoms even though lower IQ can represent a limitation in identifying obsessive thoughts. However, this aspect is not surprising since obsessive compulsive-symptoms severity generally does not correlate with IQ, as reported in a recent meta-analysis [40]. The psychopathological features impacted more on the global functioning in subjects with KBG syndrome when lower IQ was detected.

Although OCD disorders clearly appeared to be frequent in KBG, our data suggest that other psychopathological manifestations are also present in KBG syndrome with a higher prevalence compared to the general population. These features include anxiety disorder (including separation and generalized anxiety disorder). Low and coauthors reported the evidence of anxiety in KBG syndrome; however, a structural test to diagnose the disease was not performed [8].

Of note, tic disorder, depressive disorder, or ADHD disorder were also present in several cases in the studied cohort. Equal to OCD aspects, the prevalence of ADHD in our sample appears to be 10-times greater in KBG syndrome than compared to the estimated 3% prevalence reported in an Italian sample of 6183 children and adolescent schoolchildren [34]. Previous studies have reported highly variable rates for worldwide ADHD prevalence during childhood and adolescence, ranging from 0.9% to 20%, these data raised concerns about the consistency of estimates and the validity of diagnoses [41,42,43,44,45]. Our data are in line with the report by Brancati and colleagues describing atypical behavioral in seven young subjects with KBG (age ranged from 9 to 15 years) consisting of hyperactivity, anxiety, and poor concentration [46]. Notably four patients were treated with methylphenidate. Moreover, as to the prevalence of ADHD in children and adolescents with other genetic syndromes such as 22q11.2DS aged 6–17 years (*n* = 802), the rate generally was also high (37.1% in 6–12 years; 23.9% in 13–17 year old) [47]. The importance of delineating psychopathological trajectories in genetic syndromes has been demonstrated by some studies investigating 22q11.2DS and showed a high prevalence (34%) of anxiety disorder in a large sample of 1402 individuals [46]. A deeper knowledge of these aspects is expected to be useful for clinicians to plan a more accurate follow up and to routinely introduce a psychopathological assessment in the context of this syndrome. For example, anxiety has previously been found to be one of the strongest predictors of transition into psychosis in people with 22q11DS [48].

These observations were obtained by comparing prevalence rates estimated through our small sample (as expected for a monocentric study on a rare disease) to those in the whole population, and the absence of an internal control group could be a limitation. On the other hand, we used Wilson’s procedure to compute interval confidence for prevalence rates, which is particularly appropriate for small sample sizes, and a one-sample test approach (vs. population) was applied.

## 5. Conclusions

In conclusion, OCD disorder seems to be a peculiar aspect characterizing the psychopathological profile of KBG patients, which does not seem to be related to the cognitive level. Nevertheless, a better cognitive level seems to be protective on the general functioning of the subject as measured by CGAS. Notably, in a recent study, disruption of chromatin-associated transcription machinery has been associated to the phenotypic overlap between KBG and Cornelia de Lange [49], the latter being a syndrome often characterized by similar behavioral impairment including OCD disorders and general anxiety disorders [50].

The characterization of the cognitive-behavioral phenotype associated to a genetic syndrome improves the knowledge on the specific condition by identifying the relationships between behavioral characteristics and underlying genotype, with a paramount impact on clinical practice [51]. Indeed, the identification of behavioral phenotypes has important implications for genetic counselling, diagnosis, and management of children and families with genetic syndromes.

The present profiling provides new relevant information towards the definition of a psychopathological phenotype of the KBG syndrome so that the symptomatology can be appropriately managed. A limitation of our study was the absence of a control group, the relatively small sample size, and use, in the assessment, of self-reports or reports not validated for Italian population with ID. Despite any patient in our sample not being identified with ASD using K-SADS PL or neuropsychiatric evaluation according DSM, further studies of the KBG syndrome in larger patient cohorts and administering the gold standard assessment (ADI-R, ADOS) [52,53] are needed to better understand the prevalence of ASD in the KBG syndrome. Moreover, further studies on the phenotypic spectrum of KBG syndrome and its psychopathological trajectories will allow the planning of a program of mental illness prevention focused on general functioning improvement and treatments based on symptomatology (such as cognitive behavioral therapy and parental psychoeducational support).

## Figures and Tables

**Table 1 brainsci-09-00313-t001:** Descriptive characteristics of the KBG cohort included in the study.

Measure	All Subjects
Subjects (*n*)	17
Males (*n*, %)	9 (53%)
Age mean ± SD (min–max)	12.5 ± 4.0 (7.3–23.9)
Molecularly confirmed patients (*n*) cases with microdeletion (*n*, %) *ANKRD11*, frameshift change (*n*, %) *ANKRD11*, nonsense changes (*n*, %)	17 1 (6%) 12 (71%) 4 (23%)
IQ mean ± SD (min–max)	66.0 ± 16.2 (42–94)
CYBOCS Tot median (min–max) CY-BOCS Obsessions median (min–max) CY-BOCS Compulsions median (min–max)	10 (0–30) 0 (0–14) 7 (0–16)
CGAS mean ± SD (min–max)	58.7 ± 8.8 (50–75)

*n*: number, IQ: intellectual quotient, SD: deviation standard, CGAS: Children’s Global Assessment Scale, CYBOCS: Children’s Yale–Brown Obsessive-Compulsive Scale.

**Table 2 brainsci-09-00313-t002:** *ANKDR11* molecular findings detected in the present cohort of KBG patients.

Subject	Sex	Exon	*ANKRD11* Molecular Finding	Mutation Type	Segregation
1	M	1–12	16q24.3(89283689_89572450)x1	Microdeletion	de novo
2	M	9	c.6071_6084del (p.Pro2024ArgfsTer3)	Frameshift	de novo
3	F	9	c.3770_3771delAA (p.Lys1257fsTer25)	Frameshift	de novo
4	M	9	c.5205delC (p.Val1736CysfsTer227)	Frameshift	de novo
5	M	9	c.1903_1907delAAACA (p.Lys635GlnfsTer26)	Frameshift	de novo
6	F	9	c.1385_1388delCAAA (p.Thr462LysfsTer47)	Frameshift	Father untested
7	F	9	c.2412delA (p.Glu805LysfsTer58)	Frameshift	Affected mother transmitting the variant
8	M	9	c.2175_2178delCAAA (p.Asn725LysfsTer23)	Frameshift	de novo
9	M	9	c.2398_2401delGAAA (p.Glu800AsnfsTer62)	Frameshift	de novo
10	F	9	c.4389_4390delGA (p.Lys1464ThrfsTer89)	Frameshift	de novo
11	F	9	c.1903_1907delAAACA (p.Lys635GlnfsTer26)	Frameshift	de novo
12	F	9	c.2175_2178delCAAA (p.Asn725LysfsTer23)	Frameshift	de novo
13	F	9	c.1285_1286delTC (p.Ser429GlyfsTer8)	Frameshift	de novo
14	M	9	c.7416C>G (p.Tyr2472Ter)	Nonsense	de novo
15	M	9	c.1977C>G (p.Tyr659Ter)	Nonsense	de novo
16	M	9	c.7192C>T (p.Gln2398Ter)	Nonsense	de novo
17	F	9	c.3019C>T (p.Arg1007Ter)	Nonsense	de novo

M: male, F: female.

**Table 3 brainsci-09-00313-t003:** Descriptive analysis of CBCL/6–18 in our cohort of KBG Syndrome.

Child Behavior Checklist for Ages 6–18		Mean *t*-Score	Median *t*-Score	SD	Minimum *t*-Score	Maximum *t*-Score	% Subjects within the Clinical Range	% Subjects within the Borderline Range
**Syndrome Scale**	Anxious/Depressed	61.8	59.5	11.3	50	82	37.5	6.3
	Withdrawn/Depressed	62.7	60	11.0	50	89	37.5	6.3
	Somatic Complaints	60.1	57	11.0	50	90	25.0	6.3
	Social Problems	66.2	65.5	10.3	53	100	12.5	37.5
	Thought Problems	62.0	62.5	9.3	50	83	25.0	25.0
	Attention Problems	65.0	64	8.7	52	90	31.2	12.5
	Rule-Breaking Behavior	56.6	55.5	6.4	50	70	6.3	6.3
	Aggressive Behavior	59.1	56.5	8.0	50	73	18.8	6.3
**Internalizing**, **Externalizing**, **Total Problems Scale**	Internalizing Problems	61.2	61	12.0	43	83	56.3	12.5
	Externalizing Problems	56.9	56.5	8.0	40	69	25.0	12.5
	Total Problems	62.5	61	8.7	47	84	50.0	25.0
**DSM-Oriented Scale**	Affective Problems	63.6	63	9.3	51	89	31.2	6.3
	Anxiety Problems	64.2	65	10.7	51	79	56.3	0.0
	Somatic Problems	60.2	58	11.2	50	93	25.0	6.3
	Attention Deficit/Hyperactivity Problems	61.7	60	8.2	50	74	31.2	6.3
	Oppositional Defiants Problems	57.1	53.5	6.9	50	71	6.3	12.5
	Conduct Problems	55.8	56	5.0	50	66	0.0	6.3
**2007 Scale for Boys**/**Girls**	Sluggish Cognitive Tempo	61.3	61.5	6.8	50	80	18.8	12.5
	Obsessive-Compulsive Problems	59.1	55	8.9	50	80	25.0	18.8
	Post-traumatic Stress Problems	62.8	62	8.4	50	78	31.2	12.5

SD: standard deviation; **%:** percentage.

**Table 4 brainsci-09-00313-t004:** Prevalence of psychiatric disorders in KBG subjects vs. the general population (GP).

Diagnosis	KBG *n* (%)	KBG—95% CI	GP	*p*
ADHD traits or disorder	5 (29%)	13%–53%	9.9% [34]	<0.007 *
Any anxiety disorder	9 (53%)	31%–74%	24.9 % [36]	0.008 *
OCD disorder	5 (29%)	13%–53%	4.1% [33]	<0.001 *
OCD traits ^†^	9 (53%)	31%–74%	3% [33]	<0.001 *
OCD traits ^†^ or disorder	14 (82%)	59%–94%	7.1% [33]	<0.001 *
Depressive traits ^†^ or disorder	4 (24%)	10%–47%	8.2% [36]	0.021 *
Tic Disorders	3 (17.6%)	6%–41%	6.7% [35]	0.073

*n*: number, GP: general population, 95% CI: Wilson’s 95% confidence interval; ADHD: Attention Deficit Hyperactivity Disorder, OCD: Obsessive Compulsive Disorder. *, Correlation is significant at the 0.05 level (2-tailed); ^†^, Participants with “traits of disorder” showed a sub-syndromal symptomatology (i.e., not completely but almost fulfilling DSM-IV-TR criteria for the disorder).

**Table 5 brainsci-09-00313-t005:** Prevalence of specific OCD behavioral symptoms.

	% (95% CI)
**Compulsion**	
Hoarding	65% (42–83%)
Ordering	35% (17–58%)
Checking	35% (17–58%)
Washing/Cleaning	12% (3–35%)
Superstitious	6% (1–27%)
**Obsession**	
Hoarding	41% (21–64%)
Contamination	6% (1–27%)
Superstitious Somatic	6% (1–27%) 0% (0–18%)

95% CI: Wilson’s 95% confidence interval.

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
