# Peer review of "Obsessive Compulsive Symptoms and Psychopathological Profile in Children and Adolescents with KBG Syndrome"

_brainsci, 2019, doi:10.3390/brainsci9110313_

Round 1

Reviewer 1 Report

1)the introduction provides insufficient background :

it does not include all  the relevant references about KBG syndrome especially princeps description (Herrmann et al 1975) it does not include background on autism spectrum disorder in KBG

2) there are some problems with the references

the first reference is not properly placed in the first sentence. several references are used for wrong purpose  (example : in ref 2 : there are no microdeletion in this paper, please add references of papers describing  microdeletions (Marshall et al2008 Willemsen et al2010 Sacharow et al 2012,Goldenberg et al 2016) reference of review (9) is used instead of original papers, please use  od add ref  4-5-6-7-8-12

3) content

it would have been interesting to have more information about the patients, familial cases? school? way of living for adults ?(self sufficient or not, socially protected or not, treated by medications or not, working or not, having children...) since ANKRD11 is a gene implicated in ASD content is really missing in this paper about the question of autism spectrum disorder : references are missing (clinical and molecular) introduction with knowledge about ASD in KBG or at least some elements in the discussion information about theses patients (tested or not, having criteria? showing some symptoms of ASD?) discussing about the link between obsessive symptoms, anxiety, tic disorders, psychopathologic profile as studied in CBCL and ASD (symptoms that ar consecutive of an ASD?  differentiel diagnosis, shared symptoms? comorbidity? trajectories?) and the validity of score if ASD is present; and  clinical point of view for pragmatic medical, psychologic and educational purpose for following and  therapy

4)presentation

In many sentences, the references are placed in the middle of the sentence, and in some others the references are at the end. I could not find the title for figure 1 line 211 : though instead of thought

Author Response

1) the introduction provides insufficient background :

it does not include all  the relevant references about KBG syndrome especially princeps description (Herrmann et al 1975) it does not include background on autism spectrum disorder in KBG

We thank the reviewer for this comment and we added the reference and the background on ASD in KBG in the text.

2) there are some problems with the references

the first reference is not properly placed in the first sentence. several references are used for wrong purpose  (example : in ref 2 : there are no microdeletion in this paper, please add references of papers describing  microdeletions (Marshall et al 2008 Willemsen et al 2010 Sacharow et al 2012,Goldenberg et al 2016) reference of review (9) is used instead of original papers, please use  od add ref  4-5-6-7-8-12

We thank the reviewer for this comment and we apologize for the mistakes. We modified in the text the references.

3) content

it would have been interesting to have more information about the patients, familial cases? school? way of living for adults ?(self sufficient or not, socially protected or not, treated by medications or not, working or not, having children...) since ANKRD11 is a gene implicated in ASD content is really missing in this paper about the question of autism spectrum disorder : references are missing (clinical and molecular) introduction with knowledge about ASD in KBG or at least some elements in the discussion information about theses patients (tested or not, having criteria? showing some symptoms of ASD?) discussing about the link between obsessive symptoms, anxiety, tic disorders, psychopathologic profile as studied in CBCL and ASD (symptoms that ar consecutive of an ASD?  differential diagnosis, shared symptoms? comorbidity? trajectories?) and the validity of score if ASD is present; and  clinical point of view for pragmatic medical, psychologic and educational purpose for following and  therapy

We thank the reviewer for this comment and we added in section “Materials and Methods” more information about the patients according the suggestion.. Moreover, we added reference as requested about correlation between ASD and ANKDR11 gene. However in our sample no patients had diagnosis of ASD (neither in comorbidity), as emerged by the accurate neuropsychiatric evaluation according to the DSM IV criteria performed by an expert child psychiatrist investigating psychopathological signs/psychiatric disorders according to criteria of Diagnostic and Statistical Manual of Mental Disorders fourth edition (DSM-IV). According the suggestion about psychopatological profile studied in CBCL we decided to insert a new table for a better descriptive analysis of the profile in our cohort. According our neuropsychiatric evaluation results on CBCL showed an high percentage of Anxiety problems (more than an half) while social problems, usually present in ASD profile, appeared to be less involved with only 12% of patients in clinical range. In discussion we add a comment about this.  

4)presentation

In many sentences, the references are placed in the middle of the sentence, and in some others the references are at the end. I could not find the title for figure 1 line 211 : though instead of thought

We modified the text according to the comments.

Reviewer 2 Report

This study examines 17 patients with KBG syndrome and their respective psychopathological profiles, filling an important gap in the literature.  Findings were reported based on mean scores (and corresponding CIs), and prevalence rates for the sample were compared to prevalence rates reported in the literature for the general population.  Although it is important to identify the psychopathology in these individuals, there are significant concerns about the methodology and lack of scientific control in this study.

Specifically:

It appears that this sample has an extremely low IQ (66), assuming you’re using the standard mean of “normal IQ” as 100.  How did the authors evaluate the validity of any self-reports (or reports during clinician-rated interviews), since it is likely that they would not be able to understand the measures that were used in this study?

It is unclear why the authors used a repeated measures ANOVA - wasn’t this a single timepoint study in which the measures were all administered at once?  Given that there are no “repeated” measurements in this study, a repeated measures ANOVA would not be appropriate for analyzing these data.

While it may be visually interesting to see the “peaks and valleys” of the psychopathological profile across the domains on the CBCL, it’s not entirely clear how informative these data are.  It may be more fruitful to take a descriptive approach and simply report the mean t-scores and SDs (and describe the cutoffs for elevated/subthreshold and clinical scores), which would elucidate more important information re: how they compare with the general population.

The biggest concern about this paper is the lack of controls and how the authors are drawing conclusions from these data.  Specifically, it appears the authors are looking at the prevalence rates of various disorders using their sample size of 17, and then comparing these to the prevalence rates of disorders reported in the literature (using studies that have >6000 cases).  This is problematic in various ways:

4a. How were the cases in this study recruited?  For instance, were they recruited after they presented to a psychiatric clinic seeking psychological care?  If that’s the case, they are already self-selected and have a much higher chance of presenting with psychological issues than others that were not actively treatment-seeking.

4b. The small sample size is extremely difficult to establish prevalence rates for, though I also understand that KBG syndrome is exceedingly rare.  As such, I would recommend exercising extreme caution with drawing head-to-head comparisons with prevalence rates established in the literature for >6000 healthy controls, and would instead simply frame it as having XX% of the sample presenting with XX disorder (i.e., without comparison to the general population).

It is unclear what the authors mean by “traits” when they say “OCD traits” or “tic traits.”  Also, the authors split OCD traits and OCD diagnoses, but did not do this for other disorders (e.g., tics).

Since those with KBG syndrome typically have lower IQ and may also present with comorbid ASD, it is quite difficult to differentiate between true OCD compulsions and something else (i.e., repetitive behaviors in ASD, stimming).  Given the low level of insight/low IQ, what measures did the authors take to make this clinical distinction when administering the CY-BOCS?

The authors place a lot of emphasis on the prevalence of OCD and OC symptoms (especially compulsions).  However, the mean CY-BOCS score is low, and is indeed below the recommended clinical cutoff score of 16 (as used in many RCTs for OCD).  As such, it would be helpful to clarify this finding and how it may be opposite to the authors’ emphasis on the “very high prevalence of OCD.”

The authors finding about IQ level not be reing related to OC-symptom severity is not that surprising, as FSIQ generally does not correlate with OCD symptom severity, as reported in a recent meta-analysis.  

Abramovitch, A., Anholt, G., Raveh-Gottfried, S., Hamo, N., & Abramowitz, J. S. (2018). Meta-analysis of intelligence quotient (IQ) in obsessive-compulsive disorder. Neuropsychology review, 28(1), 111-120.

Author Response

Comments and Suggestions for Authors

This study examines 17 patients with KBG syndrome and their respective psychopathological profiles, filling an important gap in the literature.  Findings were reported based on mean scores (and corresponding CIs), and prevalence rates for the sample were compared to prevalence rates reported in the literature for the general population.  Although it is important to identify the psychopathology in these individuals, there are significant concerns about the methodology and lack of scientific control in this study.

Specifically:

It appears that this sample has an extremely low IQ (66), assuming you’re using the standard mean of “normal IQ” as 100.  How did the authors evaluate the validity of any self-reports (or reports during clinician-rated interviews), since it is likely that they would not be able to understand the measures that were used in this study?

We are agree with reviewer and for this reason we add this aspect as limit in the text. However, self-reports or reports were part of a broader assessment that included an accurate neuropsychiatric evaluation according to the DSM IV criteria, and moreover included a semi-structured interview with caregivers and the child separately and together. Actually in Italy doesn’t exist specific tools for psychopatological evaluation in ID and criteria for psychiatric classification in ID are not different from general population. However, all test results were analyzed by an equipe with specific expertise with Intellectual Disabilities. Finally OCD symptoms appeared to be independent from IQ (did not correlate) and, in our sample, were frequent both in our patients with ID and without.

It is unclear why the authors used a repeated measures ANOVA - wasn’t this a single timepoint study in which the measures were all administered at once?  Given that there are no “repeated” measurements in this study, a repeated measures ANOVA would not be appropriate for analyzing these data.

Repeated measures ANOVA is the equivalent of the ANOVA, but for related, not independent groups. A “repeated measures ANOVA” is also referred to as a “within-subjects ANOVA” or “ANOVA for correlated samples”. All these names imply the nature of the repeated measures ANOVA, that of a test to detect any overall differences between related means. In this sense repeated measures ANOVA is a procedure which can be used not only to assess (within-subjects) changes across time, but even changes across domains. Of course, it is mandatory that the same and unique scale is used to score each domain. However, in order to avoid misunderstanding, we have removed this analysis because it seems more appropriate to accept the reviewer's request to make a descriptive analysis of the CBCL

While it may be visually interesting to see the “peaks and valleys” of the psychopathological profile across the domains on the CBCL, it’s not entirely clear how informative these data are.  It may be more fruitful to take a descriptive approach and simply report the mean t-scores and SDs (and describe the cutoffs for elevated/subthreshold and clinical scores), which would elucidate more important information re: how they compare with the general population.

We thank the reviewer for this comment. We fully agree that a descriptive approach would be more fruitful. We have removed figure 1 and replace it with a descriptive table of the CBCL T-scores. We added the percentage of subjects who are within clinical or borderline range in the CBCL subscale. We also modified the text.

The biggest concern about this paper is the lack of controls and how the authors are drawing conclusions from these data.  Specifically, it appears the authors are looking at the prevalence rates of various disorders using their sample size of 17, and then comparing these to the prevalence rates of disorders reported in the literature (using studies that have >6000 cases).

We fully agree with this reviewer that the lack of a control group is a weakness of our study. However our aim was not to compare KBG profile to another specific “genetic” population, but to the general population (GP). In this sense we could have recruited a GP sample, e.g. by means of case-control 1:1 design. Nonetheless, we would have obtained a very imprecise estimates of prevalences in GP, clearly affected by the small sample size. Therefore, we chose to use already published and validated prevalences, usually estimated through large (>6000) samples. In other words, while a small sample is acceptable for a specific, rare and not-well studied population, a small sample is not appropriate for the general population and the use of well-recognized studies is preferable.

This is problematic in various ways:

4a. How were the cases in this study recruited?  For instance, were they recruited after they presented to a psychiatric clinic seeking psychological care?  If that’s the case, they are already self-selected and have a much higher chance of presenting with psychological issues than others that were not actively treatment-seeking.

All patients were regularly followed at service of Medical Genetics in Bambino Gesu’ and of Gemelli’s Hospital. The patients, indipendently from the presence of psychopatological problems, were evaluated by the Child neuropsychiatric Unit with the only criteria being the presence of a diagnosis of KBG molecularly confirmed. We added in the text

4b. The small sample size is extremely difficult to establish prevalence rates for, though I also understand that KBG syndrome is exceedingly rare.  As such, I would recommend exercising extreme caution with drawing head-to-head comparisons with prevalence rates established in the literature for >6000 healthy controls, and would instead simply frame it as having XX% of the sample presenting with XX disorder (i.e., without comparison to the general population).

We acknowledge the small sample size of our KBG patients and we did not pretend to make a head-to-head comparison. Rather, we compared prevalence rates of some disorders as estimated through our sample and prevalence rates of the same disorders in the whole population. Statistically this is a classic problem of one-sample tests, in which the prevalence rates of the population is assumed to be true and not affected by sampling standard errors. In addition, the width of confidence intervals is a transparent indicator of the precision of our estimates. To also be noted that we used the Wilson’s procedure, which is particularly appropriate for small sample sizes. We think that the comparison to the general population could add some insights about the characteristic of KBG syndrome and we would like to maintain it in the paper.

It is unclear what the authors mean by “traits” when they say “OCD traits” or “tic traits.”  Also, the authors split OCD traits and OCD diagnoses, but did not do this for other disorders (e.g., tics).

We thank the reviewer and we are agree that this point need to further explanation. Participants with “Trait of disorder” showed a sub-syndromal symptomatology (i.e. NOT fulfilling DSM-IV TR criteria for any disorder). In legend of table 3 we have added this definition and we have removed from the text and table 3 definition “tics trait” since tic disorders have usually dichotomous characteristics (present or not present).

Since those with KBG syndrome typically have lower IQ and may also present with comorbid ASD, it is quite difficult to differentiate between true OCD compulsions and something else (i.e., repetitive behaviors in ASD, stimming).  Given the low level of insight/low IQ, what measures did the authors take to make this clinical distinction when administering the CY-BOCS?

We agree that comparisons between the behaviours of people with KBG and those with autism could have similarities. However despite overlap in some features all our patients with KBG were evaluated by a child neuropsychiatricist by accurate neuropsychiatric evaluation according to the DSM-IV criteria and none of patients in our sample met DSM-IV criteria for autism. Moreover, when administering CY-Bocs, our KBG patients demonstrated good communication and language skills, also in case of Intellectual Disability.

The authors place a lot of emphasis on the prevalence of OCD and OC symptoms (especially compulsions).  However, the mean CY-BOCS score is low, and is indeed below the recommended clinical cutoff score of 16 (as used in many RCTs for OCD).  As such, it would be helpful to clarify this finding and how it may be opposite to the authors’ emphasis on the “very high prevalence of OCD.”

The study finds a vulnerability to different psychopathological conditions compared to the general population with a higher prevalence of OCD (one third of cases) and OC symptoms (82% with disorder and trait) especially compulsions. In a small unselected sample, a common behavioral trend (CY-BOCS mean score is mildly clinic) appears to be detected as a vulnerability condition that can evolve towards a specific psychopathological profile.

The authors finding about IQ level not be reing related to OC-symptom severity is not that surprising, as FSIQ generally does not correlate with OCD symptom severity, as reported in a recent meta-analysis.  

Abramovitch, A., Anholt, G., Raveh-Gottfried, S., Hamo, N., & Abramowitz, J. S. (2018). Meta-analysis of intelligence quotient (IQ) in obsessive-compulsive disorder. Neuropsychology review, 28(1), 111-120.

We thanks the reviewer for the suggestion and we have commented this aspect

Round 2

Reviewer 2 Report

The authors' time and responses are appreciated.  Follow-up comments are as follows:

(1) Despite the mention numerous times of "neuropsychiatric evaluation" using DSM-IV criteria (and not DSM-5, notably), it is still unclear if there were any validated interviews that were used (e.g., MINI-KID, KSADS) to actually assess for these psychiatric disorders.  This will impact the confidence in the scientific rigor in assessing for this (and thus, the subsequent conclusions).  This will need to be noted in the limitations section, if no standardized interview measure was used.

Relatedly, with the other reviewer's comments regarding ASD assessments, the authors responded that they assessed for this comorbidity; was this done with the gold-standard ADI-R/ADOS?

(2) The use of descriptive reports of the t-scores for the CBCL subscales is much improved.

(3) Were the "subsyndromal traits" determined by the interviews using DSM-IV-TR criteria, or by another method?  More information is needed on this.

(4) My previous concern regarding the ability to truly distinguish between repetitive/self-stimming behaviors versus true OCD compulsions due to confounding ASD-like features/low insight/low IQ has not been fully addressed.  At the very least, the authors should discuss this conundrum in their discussion section.

(5) My previous concerns regarding small sample size and comparing sample percentages to "general population" percentages still stands, but the authors' justification for doing so is noted.  It would be helpful for the readers to see this justification, so it is recommended that the authors include it in the methodology or discussion section.

Author Response

(1) Despite the mention numerous times of "neuropsychiatric evaluation" using DSM-IV criteria (and not DSM-5, notably), it is still unclear if there were any validated interviews that were used (e.g., MINI-KID, KSADS) to actually assess for these psychiatric disorders.  This will impact the confidence in the scientific rigor in assessing for this (and thus, the subsequent conclusions).  This will need to be noted in the limitations section, if no standardized interview measure was used.

As mentioned in the “Material and Methods” section, we used standardized interview. In order to assess current and past features of psychopathological signs/psychiatric disorders, children and parents were interviewed by using the Schedule for Affective Disorders and Schizophrenia for School Age Children, Present and Lifetime version (KSADS PL) [27] according to DSM-IV criteria, the last version of KSADS available at the beginning of the study. Neuropsychiatric evaluation was done according DSM-IV criteria in line with administering of K-Sads.

Relatedly, with the other reviewer's comments regarding ASD assessments, the authors responded that they assessed for this comorbidity; was this done with the gold-standard ADI-R/ADOS?

We agree with the reviewer that gold standard assessment for ASD is ADI-R/ADOS, but since the absence of ASD symptoms, as emerged by neuropsychiatric evaluation according DSM, the use of ADI-R/ADOS was not required. ASD diagnosis was not evident in any patient by using standardized interview KSADS PL.

(2) The use of descriptive reports of the t-scores for the CBCL subscales is much improved.

-

(3) Were the "subsyndromal traits" determined by the interviews using DSM-IV-TR criteria, or by another method?  More information is needed on this

“Participants with “traits of disordershowed a sub-syndromal symptomatology (i.e. NOT completely but almost fulfilling DSM-IV TR criteria for the disorder)”. This sentence was added in legend of table 3.

(4) My previous concern regarding the ability to truly distinguish between repetitive/self-stimming behaviors versus true OCD compulsions due to confounding ASD-like features/low insight/low IQ has not been fully addressed.  At the very least, the authors should discuss this conundrum in their discussion section.

We do agree with the reviewer that low IQ and low insight could be considered confounding factors. We added this critical issue in discussion

For all patients in whom were identified OCD symptoms both childrens and parents described characteristics of finalized behaviors (to prevent or reduce anxiety or distress, or prevent some dreaded event or situations). What is worthy of note is that despite none of the patients met criteria for ASD, mostly in case of low IQ, ASD features and OCD compulsions could be difficult to distinguish and this may represent risk of misdiagnoses and missed diagnoses”

And in conclusion

Despite in our sample not being identified any patient with ASD by using KSADS PL or neuropsychiatric evaluation according DSM, further studies in KBG syndrome in larger patient cohorts and administering gold standard assessment (ADI-r, ADOS) are needed to better understand prevalence of ASD in KBG syndrome”.

(5) My previous concerns regarding small sample size and comparing sample percentages to "general population" percentages still stands, but the authors' justification for doing so is noted.  It would be helpful for the readers to see this justification, so it is recommended that the authors include it in the methodology or discussion section.

According to the Reviewer’s suggestion and to inform readers about the approach of comparing prevalence rates (estimated in our small sample) to those of general population, a sentence was added in the discussion and the lack of a control group as possible study limitation.